# The Perceived Ability of Gastroenterologists, Hepatologists and Surgeons Can Bias Medical Decision Making

**DOI:** 10.3390/ijerph17031058

**Published:** 2020-02-07

**Authors:** Alessandro Cucchetti, Dylan Evans, Andrea Casadei-Gardini, Fabio Piscaglia, Lorenzo Maroni, Federica Odaldi, Giorgio Ercolani

**Affiliations:** 1Department of Medical and Surgical Sciences-DIMEC, S.Orsola-Malpighi Hospital, Alma Mater Studiorum-University of Bologna, 40138 Bologna, Italy; aleqko@libero.it (A.C.); fabio.piscaglia@unibo.it (F.P.); lorenzo.maroni@studio.unibo.it (L.M.); federica.odaldi@studio.unibo.it (F.O.); giorgio.ercolani2@unibo.it (G.E.); 2Morgagni–Pierantoni Hospital, 47121 Forlì, Italy; 3Projection Point, T23 Cork, Ireland; evansd66@googlemail.com; 4Division of Medical Oncology, Department of Medical and Surgical Sciences for Children and Adults, University Hospital of Modena, Via Del pozzo n 71, 41122 Modena, Italy

**Keywords:** Dunning–Kruger, medical decision making, medical error, survey, hepatology, surgery

## Abstract

Medical errors are a troubling issue and physicians should be careful to scrutinize their own decisions, remaining open to the possibility that they may be wrong. Even so, doctors may still be overconfident. A survey was here conducted to test how medical experience and self-confidence can affect physicians working in the specific clinical area. Potential participants were contacted through personalized emails and invited to contribute to the survey. The “risk-intelligence” test consists of 50 statements about general knowledge in which participants were asked to indicate how likely they thought that each statement was true or false. The risk-intelligence quotient (RQ), a measure of self-confidence, varies between 0 and 100. The higher the RQ score, the better the confidence in personal knowledge. To allow for a representation of 1000 physicians, the sample size was calculated as 278 respondents. A total of 1334 individual emails were sent to reach 278 respondents. A control group of 198 medical students were also invited, of them, 54 responded to the survey. The mean RQ (SD)of physicians was 61.1 (11.4) and that of students was 52.6 (9.9). Assuming age as indicator of knowledge, it was observed that physicians ≤34 years had a mean RQ of 59.1 (10.1); those of 35–42 years had 61.0 (11.0); in those of 43–51 years increased to 62.9 (12.2); reached a plateau of 63.0 (11.5) between 52–59 years and decreased to 59.6 (12.1) in respondents ≥60 years (r^2^:0.992). Doctors overestimate smaller probabilities and under-estimate higher probabilities. Specialists in gastroenterology and hepato-biliary diseases suffer from some degree of self-confidence bias, potentially leading to medical errors. Approaches aimed at ameliorating the self-judgment should be promoted more widely in medical education.

## 1. Introduction

William Osler famously remarked that “medicine is a science of uncertainty and an art of probability”. No doctor returns home from a hard day at the hospital without the nagging feeling that some of his/her diagnoses may turn out to be wrong or that some treatments may not lead to the expected result. Even so, doctors may still be overconfident. That is, doctors may assume that their diagnoses and prognoses are significantly more accurate than they really are. Psychologists refer to such illusory superiority as the Dunning–Kruger effect, a meta-cognitive bias that leads to a discrepancy between the way people actually perform and the way they perceive their own performance [1]. 

Over decades of research, involving a wide variety of domains and circumstances, psychologists have examined how accurately people judge themselves [2]. The typical finding is that people have a flawed self-assessment with negligible correlations between skill perception and objective performance. In addition, people tend to be too optimistic about their expertise, knowledge and future prospects. There are scant data regarding self-confidence and self-judgment of physicians involved in clinical settings, and unfortunately the few available studies yielded very pessimistic conclusions [3,4,5]. Given the serious and potentially fatal consequences of poor performance in medicine due to erroneous self-confidence, the lack of research in this area is worrisome. Gastroenterologists, hepatologists, as well as specialists in infectious disease, internal medicine and hepato-pancreato-biliary (HPB) surgeons, are daily involved in taking difficult clinical decisions involving a large part of the population. In 2015, the World Health Organization estimated that 325 million people were living with chronic hepatitis infections worldwide from hepatitis B virus and hepatitis C virus and that, globally, 1.34 million people died of the virus [6]. The global burden of hepatitis is forecast to grow over the next two decades [6,7,8]. Complications related to this disease, as well as alcohol-related and non-alcoholic liver diseases, are the main reasons for seeking the advice of specialists in this area [6,7,8]. In addition, hepatocellular carcinoma, the fifth most common cancer in men and the ninth in women, arises on this ground, often requiring surgical referral for evaluation [7,8].

Poor diagnoses as well as over-confident prognoses can lead to significant payoffs, both in financial terms and in terms of human suffering [9,10]. The discrepancy between the way physicians actually perform and the way they perceive their own performance can be measured by a test defined as “risk-intelligence” test, which is based on questions of common knowledge [11]. This test essentially reflects “a special kind of intelligence for thinking about risk and uncertainty”, at the core of which is the ability to estimate probabilities according to the self-confidence of respondents [11]. As a step towards remedying these deficits, we carried out a study to test how medical experience and self-confidence, assessed through risk-intelligence test, can affect medical decision-making in a specific area of clinical medicine, namely gastroenterology and related specialties.

## 2. Methods

### 2.1. Study Design

The present survey was designed to test the hypothesis that self-judgment varies with the extent of professional experience, here measured through age, that is, how the increased amount of experience could modify self-confidence in personal knowledge. The design was developed on the basis of information from a single focus group of physicians, including a leading psychologist, experts in methodology of clinical trials, and a review of the available literature [2,11]. The outcome measure was the “risk-intelligence” quotient (RQ) for each respondent as assessed by the dedicated test. Selected participants were required to have a university degree in medicine, a requirement that was indirectly confirmed by asking participants to identify their specialty. As already used in pertinent literature, the participants’ age was taken as proxy for experience [12,13].

Potential participants were selected in the field of medical or surgical expertise of the authors, namely gastroenterology and hepato-biliary diseases, then, contacted through a personalized email and invited to contribute to the survey. A web-link was provided with the email. The name and email of each potential respondent was retrieved by an extensive search through abstract books of national (Italian Association for the Study of the Liver—AISF) and international (European Association for the Study of the Liver—EASL; American Association for the Study of Liver Diseases—AASLD) congresses of the last five years, Medline research, single institutional, hospital and academic websites, personal websites, scientific journals and other free internet resources, in order to identify specialists involved in the specific fields of gastroenterology and hepatology, including specialists in internal medicine and infectious disease, and HPB surgeons who have “hepatic disorders” as one of the main field of interest/expertise.

Between May 2017 and September 2018, identified physicians received a personalized e-mail describing the nature of the study, its concept and the link to the online test. After a first call, non-respondents were contacted by e-mail through a reminder sent after 7 days and then after 14 days. A final call was re-sent to all retrieved potential participants when the 90% of the sample was achieved. Physicians who did not respond at the end of the study were considered as non-responders.

An additional group of medical students participated obtained outside the regular lesson hours, in a completely voluntary manner and informing students that their participation was completely anonymous and would not have modified the profit examination in any way. This group was chosen because well fit the starting hypothesis that self-judgment varies with the extent of experience measured through age. 

The study was approved by the Local Ethics Committees (Approval n. 105543).

### 2.2. Survey Design

The “risk-intelligence” test consists of 50 statements about general knowledge, which may be true or false [11]. The participants were asked to indicate how likely they thought that each statement was true or false. The complete list of statement used is reported in the Table 1. 

As an example: “the Euphrates river runs through Baghdad”—if participants were absolutely sure that a statement was true, they were instructed to click on the button marked 100%. Conversely, if participants were completely convinced that a statement was false, they were instructed to click on the button marked 0%. In cases where participant was not fully convinced whether the statement was true or false, they were instructed to click on the buttons marked from 10% to 90%, depending on how sure they were. That is, if participants thought that the statement was probably true, but they were not absolutely sure, they were instructed to click on the 60%, 70%, 80%, or 90% buttons, depending on how sure they were, and, similarly, if participants thought that the statement was probably false, but they were not absolutely sure, they were instructed to click on the 40%, 30%, 20%, or 10% buttons. If participants were completely unaware or uncertain as to whether the statement was true or false, they were asked to click the 50% button. A version of the test can be found and tried at: https://www.projectionpoint.com/index.php.

All available data were collected in an Excel spreadsheet and RQ values properly calculated. The risk intelligence quotient (RQ) is a number between 0 and 100. The higher the RQ score, the higher is the ability to make accurate probability estimates, reflecting adequate self-confidence in own knowledge. In other words, higher scores are provided by least discrepancy between the real outcome and the self-perceived knowledge. The risk score drops to minimal when the self-perceived knowledge is wrong (e.g., 100% certainty of true statement for a false statement or vice versa). An uncertain answer (e.g., 60% of true statement) would less heavily influence the outcome in case the judgment was eventually wrong (i.e., in case of real false statement) than a certainty (100%) of true statement. In simple words, a cautious approach is preferred (higher scores) to a wrong self-confidence, but a correct self-confidence, when possible, is obviously best. Mathematical details regarding the calculation of RQ scores are reported in the Table 2. 

Finally, participants were asked to give detail about demographical and specialty data. 

All participants were informed that only the principal investigator had full access to all data and that the survey was completely anonymous. Participants were also informed that by accepting the invitation, and completing the task, they gave their approval to the use of their data. The survey was completely unfounded and there was no remuneration for respondents. 

### 2.3. Survey Sample

To allow for a representation of 1000 physicians with a margin of error of 5% and a confidence level of 95%, the calculated sample size was determined as at least 278 respondents [14]. Because we were unaware of the potential response rate, an iterative approach was adopted to reach the planned sample size. Thus, we progressively increased the number of individuals contacted until the planned sample size was reached. Consequently, the response rate was calculated post-hoc.

### 2.4. Statistical Analysis

All categorical variables are reported as number of subjects and frequencies; continuous variables are described as means and standard deviations or standard errors. Since, as previously stated, the present sample was representative of a larger population of physicians, the effect size was preferred to *p*-value calculation since it is independent from the size of the study group [15,16]. The Cohen’s *d*-value was consequently applied and interpreted as previously suggested: *d*-values <|0.1| indicated negligible differences between the means; *d*-values between |0.1| and |0.3| indicated small differences, *d*-values between |0.3| and |0.5| indicated moderate differences, and *d*-values >|0.5| indicated considerable differences. In addition, the Cohen’s U_3_ was calculated: in the present study, this measure indicated the proportion (percentage) of respondents in a specific age class who will have a RQ above the mean of another age class [16].

## 3. Results

A total of 1334 individual emails were sent and a total of 285 subjects participated in the survey. Seven responses were excluded because the declared specialty did not fit the aim of the sample size (one specialist in public health, one in epigenetics, one biologist, one epidemiologist, one specialist in critical care, and three who did not declare their specialty). The planned sample of 278 respondents was achieved by 27 September 2018, then the survey closed (response rate: 20.8%). In the control group, a total of 198 medical students were invited, and 54 responded to the survey (response rate: 27.3%).

Respondent physicians declared their specialty as follows: hepatology n: 79, (28.4%); gastroenterology, n: 78 (28.1%); internal medicine, n: 55, (19.8%); surgery, n: 46 (16.5%); infectious disease, n: 19 (6.8%) and geriatrics, n: 1 (0.4%). The majority of respondent physicians were male (n: 181; 65.1%), the mean age was 47.1 ± 12.2 years (median: 46; range: 26–74 years). The following age quintiles were consequently identified: ≤34 years (n: 58), 35–42 years (n: 57), 43–51 years (n: 56), 52–59 years (n: 54) and ≥60 years (n: 53). In the control group of 54 medical students, the majority were females (n: 44; 81.5%) and the mean age was 22.7 ± 2.1 years (median: 22; range: 21–33 years).

### 3.1. Risk-Intelligence Results

The mean RQ of participant physicians was 61.1 ± 11.4 (median: 61.0; range: 30.7–88.7; Figure 1). 

The mean RQ of medical students was 52.6 ± 9.9 (median: 54.2, range: 33.6–75.3). Thus, respondent physicians showed an average RQ higher than that of medical students (*d* = 0.796) and this difference is interpretable as that 79% of physicians will have an RQ above the mean of that of medical students (U_3_ value).

The relationship between RQ and age was then explored between the quintiles identified (Figure 2).

In physicians ≤34 years the mean RQ was 59.1 ± 10.1; in those of 35–42 years it was 61.0 ± 11.0; in those of 43–51 years increased to 62.9 ± 12.2; reached a plateau of 63.0 ± 11.5 in respondents of 52–59 years and decreased to 59.6 ± 12.1 in respondents ≥60 years. 

In practical terms, 73% of physicians aged ≤34 years would have an average RQ above that of medical students and 57% of respondents aged 35–42 years would have a RQ above the mean of younger physicians (≤34 years). With aging this percentage increased up to 64% but, finally, the trend was reversed and 61% of physicians ≥60 years would have a RQ below the mean of respondents of 52–59 years. The curve was fitted with a polynomial regression line with an r^2^ of 0.992 (Figure 2).

### 3.2. Calibration Curve

Calibration curve represents a way to visualize RQ scores. The average calibration curve of respondent physicians is depicted in the Figure 3. 

A perfect calibration curve would lie on the identity line x = y. The further away from that diagonal line the curve lies, the poorer the calibration is. As can be noted, for probabilities <60% the perceived probability was consistently higher than the actual, meaning that the physicians over-estimated low probabilities. On the contrary, above this threshold, they tend to under-estimate probabilities. 

## 4. Discussion

It was recently reported that if medical error were a disease, it would rank as the third leading cause of death in the United States immediately after cardiovascular disease and cancer [10]. Physicians should therefore be careful to scrutinize their own decisions, and remain open to the possibility that they may be wrong. Furthermore, yet there is evidence that those who are most likely to make mistakes are also the least likely to recognize their errors. As Dunning and Kruger argued, when people are unskilled, they suffer a dual burden: not only they reach erroneous conclusions and make bad choices, but their inability robs them the capacity to realize the problems [1,2,17]. Although they make mistakes, they are left with the impression that they are doing just fine as Charles Darwin wisely noted over a century ago, that “ignorance more frequently begets confidence than does knowledge” [18].

Overconfidence is one of the most significant cognitive biases and people usually overestimate themselves [2]. They hold over-inflated views of their experience and talents. It is in the health domain that divergences between self-perceptions of knowledge and reality have been most commonly documented. As some classical examples: adolescent boys’ confidence in their knowledge about how to use condoms correlates only marginally with their actual knowledge [18]. The nurses’ confidence in their knowledge of basic life-support tasks fails to correlate at all with their actual level of knowledge [19]. Physicians’ self-rated knowledge about thyroid disorders also fails to correlate with their performance on a quiz on the topic [3]. Surgical residents’ views of their surgical ability failed to correlate with their true performance at a standardized exam [4]. The most sobering finding reported in literature is that doctors asked for pneumonia diagnosis estimated a 90% chance that their patients suffered from it but only about 15% of the patients turned out to truly have the disease [5]. In other words, people as well as doctors have a warp view of their expertise, skill and knowledge resulting in supposed performance and accuracy not supported by the evidence. This meant that physicians were likely to recommend more tests than those strictly necessary, prescribe more treatments than warranted, and also cause their patients needless worry [9,10]. 

In the present study, we investigated how over-confidence affects clinicians belonging to a particular clinical area, an area having a great importance in the global health system considering the worldwide burden of liver diseases [6,7,8]. By using age as indicator of a physician’s experience we showed that younger and older doctors were the most affected by the self-confidence bias, and that best clinical performances can be achieved by middle aged doctors (Figure 2). Our results suggest that novice doctors overestimate their competence and this observation is reinforced by findings in the control group of medical students who showed the lowest RQ scores. As time goes by, doctors begin to see how little they actually know, returning more appropriate self-confidence and ultimately more accurate forecasts. The more experienced physicians in our study seem better able to handle doubts and uncertainties, as well as relevant and useful information that they do have in hand. Yet, this progress is not monotonic, since after a certain age the RQ scores drop to values near to those of novices. This drop in the scores of older physicians may be due to a concept arising from the feeling that one has learned everything needed to know about a particular clinical area. Several psychological mechanisms collude to produce these faulty self-assessments, but most of them can be summarized into two general categories. At first, erroneous self-assessments arise because medical students and young doctors do not have all the information necessary to provide accurate assessments, and obviously they do not take into account what they do not know [2]. Then, doctors progressively acquire information and learn that medicine is more complicated than it seems, developing the necessary criticism in their own knowledge to produce more accurate forecasts. After a certain age, the second erroneous self-assessment is likely arising when older physicians proudly neglect novel relevant, useful information that they do have at hand, thus returning low RQ scores [2]. It should be pointed out that the lower RQ scores of older doctors might also be due to age-related decline in various aspects of cognition. However, the older respondents in our study were between 60 and 74 years old, and it seems premature to suppose a loss of other cognitive abilities [12,13].

We have already examined the importance of cognitive biases in the specific field of gastroenterology and hepatology decision-making in the past [20]. In particular, we showed how surgeons and hepatologists can propose either surgical or non-surgical therapies for the same patients, suffering from hepatocellular carcinoma, in a setting analogous to the present one, on the basis of their perceived regret [20,21,22,23]. With the present results we add another significant piece of decision-making cognitive process in the specific clinical scenario in which we daily move. However, how we can escape from these psychological traps to provide the best decision-making for patients?

We believe that we can move through two different levels, one based on interventions aimed at encouraging the comparison between different specialists and one based on interventions aimed at reducing unrealistic optimism and/or pessimism of physicians leading to over-estimation and/or under-estimation of probabilities (Figure 3) [2].

The appraisal of different specialists in a specific clinical condition can lead to the optimization of the decision-making, considering different point-of-views, clinical feelings and judgments of different specialists, sharing the final decision. The multidisciplinary approach should lead each specialist to learn not only new information, but also to calibrate his/her own optimism (and pessimism) ameliorating self-confidence. In the second level, specialists should obtain a feedback of their own decisions and involves medical teaching. Some medical schools are beginning to use “confidence-based assessment” or “certainty-based marking” approach [23,24,25,26]. In this form of assessment, medical students must not only give the right answer but also assess the confidence with which they give each answer. If students give the wrong answer confidently, they receive the worst possible grade. If they give the wrong answer but are not confident, they get a better grade. Giving the right answer without confidence is okay but not ideal, as in real life it could end up with their wasting time having to consult others. The best answer is that which is correct and made with confidence. This form of assessment is a very effective way both to highlight the need to assess one’s degree of confidence in an analysis of the available information and to provide feedback [23,24,25,26]. This approach should be also applied to doctors’ performance evaluation to improve self-confidence and decision-making.

We should however hasten to add that the present study analyzes only one aspect of all known cognitive biases affecting medical decision-making and that there are other psychological mechanisms responsible for errors in self-assessment and medical tasks [2]. Literature abounds with demonstrations of all the techniques people utilize to construct and maintain desirable images of themselves while avoiding negative ones [27,28] but some aspects of these demonstrations are also present in the material we discuss here. Thus, it should be understood that there are other processes that might lead doctors to form incorrect diagnosis and medical decisions. Additionally, it is necessary to emphasize that the RQ test was developed using general questions, rather than medical questions, to allow for its utilization in different group of subjects as, for example, in professional gamblers and in weather-forecasters [11]. How the RQ test results could change using specific medical question is a question mark which could be explored with future, dedicated studies.

## 5. Conclusions

In conclusion, we showed that specialists in the fields of gastroenterology and hepato-biliary disease suffer from some degree of Dunning–Kruger effect as any other person would do. Approaches aimed at ameliorating the self-judgment should be promoted more widely in medical education and even during daily clinical practice because overconfidence is an important, though neglected source of medical errors. Greater efforts need to be produced to correct it.

## Figures and Tables

**Figure 1 ijerph-17-01058-f001:**
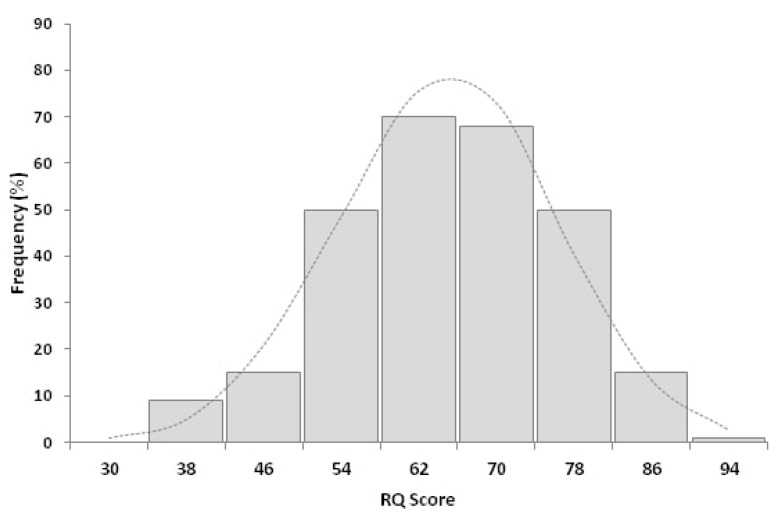
Distribution of risk-intelligence quotient (RQ) among 278 respondents to the survey. The mean RQ was 61.1 ± 11.4, the median was 61.0 (range: 30.7 and 88.7).

**Figure 2 ijerph-17-01058-f002:**
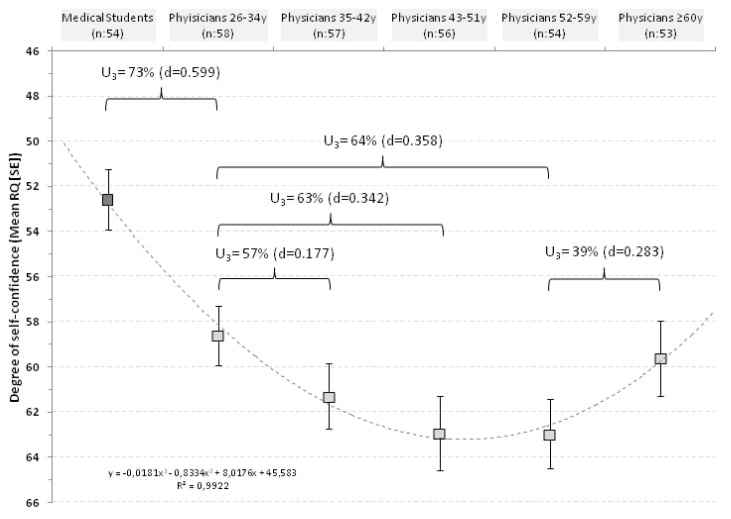
Distribution of RQ score in relationship with age classes in physicians and in medical students.

**Figure 3 ijerph-17-01058-f003:**
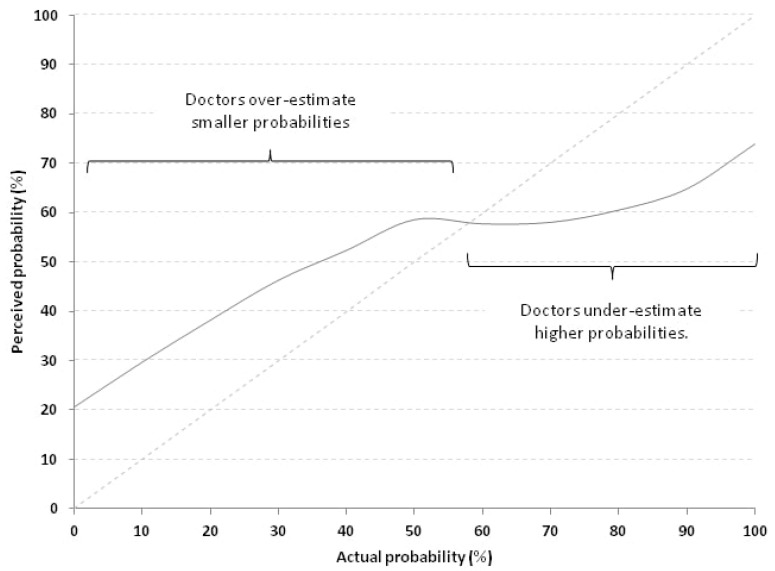
Calibration curve of all 278 physicians which compares perceived probabilities (continuous grey line) with actual probabilities (dotted line lying on the identity line x = y). Doctors overestimate smaller probabilities and under-estimate higher probabilities.

**Table 1 ijerph-17-01058-t001:** Statements used in the present risk-intelligence quotient (RQ) test. The respondents did not have access to the correct answers, reported here only for completeness of the information.

	Correct Answer
Q1: The Euphrates river runs through Baghdad	FALSE
Q2: Norway is often called the “land of the midnight sun”	TRUE
Q3: The world’s windiest place is Chicago.	FALSE
Q4: Canberra is the capital of Australia.	TRUE
Q5: The only capital city that stands on the river Danube is Belgrade.	FALSE
Q6: Africa is the largest continent.	FALSE
Q7: There are more people in the world than chickens.	FALSE
Q8: Greenland is larger than Australia.	FALSE
Q9: The San Andreas Fault forms the tectonic boundary between the Pacific Plate and the North American Plate.	TRUE
Q10: Velvet bushes are native to Australia.	TRUE
Q1: A one followed by 100 zeros is a Googol.	TRUE
Q2: The ‘Spanish Flu’ killed more people in the 1918–1919 world-wide pandemic than did the First World War.	TRUE
Q3: The most frequently diagnosed cancer in men is prostate cancer.	FALSE
Q4: Lightning kills fewer than 500 people per year.	FALSE
Q5: Mercury is the only planet within our solar system that rotates clockwise.	FALSE
Q6: It is possible to lead a cow upstairs but not downstairs, because a cows’ knees can’t bend properly to walk back down.	TRUE
Q7: US Dollar bills are made out of cotton and linen.	TRUE
Q8: The Earth is older than the moon.	TRUE
Q9: Honeybees never sleep.	FALSE
Q10: Iron accounts for over 30 percent of the mass of the Earth.	TRUE
Q1: The industrial revolution first took place in the United States.	FALSE
Q2: The last Inca emperor was Montezuma.	FALSE
Q3: In Roman mythology, Mars was the god of war.	TRUE
Q4: Alfred Nobel invented dynamite before 1800.	FALSE
Q5: Mao Zedong declared the founding of the People’s Republic of China in 1949.	TRUE
Q6: Over 40% of all deaths from natural disasters from 1945 to 1986 were caused by earthquakes.	TRUE
Q7: The Taj Mahal was built by Emperor Shah Jahan in memory of his favourite wife.	TRUE
Q8: Christianity became the official religion of the Roman empire in the third century AD.	FALSE
Q9: The first democrat president of America was Abraham Lincoln.	FALSE
Q10: The great pyramid of Giza was built more than 5000 years ago.	FALSE
Q1: In 1994, Bill Clinton was accused of sexual harassment by a woman called Paula Jones.	TRUE
Q2: Lehman Brothers went bankrupt in September 2008.	TRUE
Q3: More than 10 American states let citizens smoke marijuana for medical reasons.	TRUE
Q4: The Islamic Resistance Movement is better known to Palestinians as Hizbollah.	FALSE
Q5: Wikipedia was launched in 1999 by Jimmy Wales and Larry Sanger.	FALSE
Q6: There have been over 40 US presidents.	TRUE
Q7: Most of the terrorists who carried out the attacks on 9/11 were from Saudi Arabia.	TRUE
Q8: Over 50% of Nigeria’s population lives on less than one dollar per day.	TRUE
Q9: China has a greater gross domestic product than Japan.	TRUE
Q10: Dmitry Medvedev was the successor of Vladimir Putin as President of Russia.	TRUE
Q1: Zinedine Yazid Zidane played on the French national team for over 5 years.	TRUE
Q2: Harry Potter and the Goblet of Fire tells the story of Harry Potter’s third year at Hogwarts.	FALSE
Q3: Lauren Bacall was Humphrey Bogart’s third wife.	FALSE
Q4: LL Cool J got his name from the observation “Ladies Love Cool James”.	TRUE
Q5: Germany hosted the football World Cup in 2008.	FALSE
Q6: Male gymnasts refer to the pommel horse as “the pig”.	TRUE
Q7: The word ‘robot’ was coined by the American science fiction writer, Isaac Asimov.	FALSE
Q8: Russia has more movie theatres than any other country.	TRUE
Q9: China has more English speakers than the United States.	FALSE
Q10: Cel animation is a form of 2D animation.	TRUE

**Table 2 ijerph-17-01058-t002:** Spreadsheet Showing How RQ Scores Are Calculated.

(A) Category	(B) Estimates	(C) True	(D) Percent True	(E) Residuals (*R*)	(F) 100–*R*	Column F × Column B
0	10	1	10	10	90	900
10	10	1	10	0	100	1000
20	10	2	20	0	100	1000
30	10	4	40	10	90	900
40	10	4	40	0	100	1000
50	10	5	50	0	100	1000
60	10	7	70	10	90	900
70	10	7	70	0	100	1000
80	10	6	60	20	80	800
90	10	8	80	10	90	900
100	10	8	80	20	80	800
Total	110					10,200
Weighted mean						93
RQ score						86

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
