# Peer review of "The Perceived Ability of Gastroenterologists, Hepatologists and Surgeons Can Bias Medical Decision Making"

_ijerph, 2020, doi:10.3390/ijerph17031058_

Round 1

Reviewer 1 Report

The manuscript is interesting however there are a number of issues to revise. These are indicated in the pdf document attached. Please refer to a checklist as you report your findings. 

The references should be within the sentence not outside it. Please correct this throughout the manuscript.

Author Response

Answers to Reviewer #1

The manuscript is interesting however there are a number of issues to revise. These are indicated in the pdf document attached. Please refer to a checklist as you report your findings. 

The references should be within the sentence not outside it. Please correct this throughout the manuscript. Line 61: Move this to before the full stop; please do this for all your reference.

Re: We have moved all the references before the full stop.

Line 72: Please revise the entire methods section. It is not clear and concise. Can one repeat the study using your description here?

Re: We have revised all methods section. We hope that is more clear respect the previous version. We feel that methodology is reproducible on different groups of participants.

Line 81: Please include the selection criteria more so the exclusion criteria. Please describe the participants/recruitment in more detail and in one section instead of including it in the study design and survey design sections.

Re: We followed Reviewer suggestion in the present version of the manuscript.

Line 83: Please provide more information on the control group…was this the best group to use as a control group? Why didn’t you compare these specialists to another specialty instead of medical students who are not experienced medical personnel?

Re: Reviewer raised an interesting question. We do not know if medical students can represent the best group to use as control. However, the concept of risk-intelligence does not rely on the knowledge by itself but on the confidence that each participant has in his/her own knowledge. Thus, medical students well represent a group of subjects which had, for definition, less knowledge than MD responders, fulfilling the initial requirement to test the hypothesis that self-judgment varies with the extent of experience, here measured through age, that is, how the increased amount of experience could modify self-confidence in personal knowledge. We briefly added this specification in the method section.

Lines 85-86: This should be included in the participant recruitment section and not here

Re: We agree with the reviewer and we have revised all methods section.

Line 90: Capitalize T in the table and is that the correct table number-S1; please revise this sentence

Re: We modified accordingly. We apologize for this mistake.

Line 91 table 1: Please label the table correctly. Is this a sample of the statements and the correct answer? There are statements and not questions. They have the answers too-marking sheet?

Re: We labeled table 1 as appropriate. These are all the statements used (and not a sample) for RQ test, and participants did not have correct answers (as stated, the survey was sent through a google form so that correct answers were not available). We thus modified the title of the table accordingly.

Line 216: Perhaps you could have included the first two paragraphs in the introduction section instead of here. The first paragraph ought to have been summary of the main key findings of the paper

Re:We revised all methods section. We hope that is more clear respect the previous version.

Line 260: Perhaps this theory could have been introduced in the introduction section and the methods section as there is no theoretical background indicated

Re: Regret theory represent a potential cognitive bias in the decision making of patients in the field of hepatology. We added some specification about this in the discussion section to corroborate results from present study’s results on another potential cognitive bias, namely “Dunning-Kruger” effect. However, we agree that the paragraph is actually too long and is not fully appropriate. It would not even be appropriate for the introduction, so we consistently summarized that paragraph in the present revision.  

Line 264: Please revise this; Line 265: please revise this sentence

Re: we consistently summarized this part of the manuscript

Line 276: reference?

Re: We added as appropriate

Reviewer 2 Report

Dear authors,

congratulations on your paper. The referred thematic is totally relevant since physicians represent a critical profession that can save people. A simple error can represent a horrible consequence. I have few considerations about the paper.

At lines 56-58, you cited "chronic hepatitis infections" and after you exposed the number of people who died from the virus. What virus? Because, for example, hepatitis B virus (HBV) and hepatitis C virus (HCV) are virus that can cause chronic hepatitis; The name of tables are different between text and legend (Example: text: Table S1 and legend: Table 1); I understood the metodology named RQ test, but I was thinking about the relevance of using generic questions, such as “The Euphrates river runs through Baghdad” or “The world's windiest place is Chicago”. In my opinion, the test should priorize medical questions. I understood that generic questions were included in test to analyze self-confidence in comparison to experience but I think that they do not reflect real medical routine. This is not a negative comment. I only  highlighted this topic for you to think about how future tests should be made to test physicians experience in their specific fields.  I judged worrisome that older doctors were one of the groups that were most affected by the self-confidence bias. For future studies, you should priorize that group (≥60 years) since many patients search for more "experienced" physicians because they believe it's the best option.

Author Response

Answers to Reviewer #2

Dear authors, congratulations on your paper. The referred thematic is totally relevant since physicians represent a critical profession that can save people. A simple error can represent a horrible consequence. I have few considerations about the paper.

At lines 56-58, you cited "chronic hepatitis infections" and after you exposed the number of people who died from the virus. What virus? Because, for example, hepatitis B virus (HBV) and hepatitis C virus (HCV) are virus that can cause chronic hepatitis.

Re:That’s right and we add in the text “hepatitis B” virus and “hepatitis C” virus.

The name of tables are different between text and legend (Example: text: Table S1 and legend: Table 1)

Re:We now corrected this error. We apologize for this mistake.

I understood the methodology named RQ test, but I was thinking about the relevance of using generic questions, such as “The Euphrates river runs through Baghdad” or “The world's windiest place is Chicago”. In my opinion, the test should priorize medical questions. I understood that generic questions were included in test to analyze self-confidence in comparison to experience but I think that they do not reflect real medical routine. This is not a negative comment. I only highlighted this topic for you to think about how future tests should be made to test physicians experience in their specific fields.

Re:This is a very pertinent and intriguing observation. The RQ test was developed using general question to allow for its utilization in different group of subjects. For example, it was already tested in professional gamblers and in weather-forecasters [Evans D. Risk Intelligence: How to Live with Uncertainty. Atlantic Books 2012, London, UK. ISBN: 1848877382.]. This is the reason why general and not medical questions were used, to provide an eventual comparison between different professional. We added some discussion in the last paragraph of the discussion section. We thank the Reviewer for this suggestion.

I judged worrisome that older doctors were one of the groups that were most affected by the self-confidence bias. For future studies, you should priorize that group (≥60 years) since many patients search for more "experienced" physicians because they believe it's the best option.

Re:Reviewer hit the target! This is why some of our older colleagues do not appreciate the result of this study! We will take into consideration this suggestion for future studies on this topic.

Round 2

Reviewer 2 Report

Dear authors,

I appreciate your responses and feel satisfied with modifications. Congratulations on your paper.